# Socioeconomic status and pulmonary function, transition from childhood to adulthood: cross-sectional results from the polish part of the HAPIEE study

Maciej Polak,[1] Krystyna Szafraniec,[1] Magdalena Kozela,[1] Renata Wolfshaut-Wolak,[1] Martin Bobak,[2] Andrzej Pająk[1]

[1]Department of Epidemiology and Population Studies, Jagiellonian University Medical College, Krakow, Poland
[2]Department of Epidemiology and Public Health, University College London, London, UK

**Correspondence to**
Dr Maciej Polak;
maciej.1.polak@uj.edu.pl

## ABSTRACT

**Objective** Previous studies have reported inverse associations between socioeconomic status (SES) and lung function, but less is known about whether pulmonary function is affected by SES changes. We aimed to describe the relationship of changes of SES between childhood and adulthood with pulmonary function.

**Design** Cross-sectional study.

**Participants** The study sample included 4104 men and women, aged 45–69 years, residents of Krakow, participating in the Polish part of the Health, Alcohol and Psychosocial Factors in Eastern Europe Project.

**Main outcome** Forced expiratory volume ($FEV_1$) and forced vital capacity (FVC) were assessed by the standardised spirometry procedure. Participants were classified into three categories of SES (low, moderate or high) based on information on parent's education, housing standard during childhood, own education, employment status, household amenities and financial status.

**Results** The adjusted difference in mean FVC between persons with low and high adulthood SES was 100 mL (p=0.005) in men and 100 mL (p<0.001) in women; the differences in mean $FEV_1$ were 103 mL (p<0.001) and 80 mL (p<0.001), respectively. Upward social mobility and moderate or high SES at both childhood and adulthood were related to significantly higher $FEV_1$ and FVC compared with low SES at both childhood and adulthood or downward social mobility.

**Conclusions** Low SES over a life course was associated with the lowest lung function. Downward social mobility was associated with a poorer pulmonary function, while upward mobility or life course and moderate or high SES were associated with a better pulmonary function.

## INTRODUCTION

Socioeconomic status (SES) is one of the main predictors of health at each stage of the life cycle.[1–3] People in lower socioeconomic groups usually experience poorer health than people from higher socioeconomic groups, regardless of the health measures and SES indicators used.[4–6] SES is a complex, multidimensional social construct conceptualised to include income, education and occupation

### Strengths and limitations of this study

► This is a first study addressing the association between social mobility and pulmonary function in single urban population in Central Eastern Europe.
► The study sample was selected from the general population aged 45–69.
► Multidimensional nature of socioeconomic status (SES) was explored.
► The cross-sectional study design does not allow to address the causality of the observed relations.
► Distant recall of the markers of social circumstances could have contributed to overestimation of childhood SES.

(and other factors) which are often interrelated. However, it is likely that each SES measure reflects slightly different forces associated with health and disease on individual and societal levels.[7]

Several social, psychological and biological mechanisms have been proposed to explain the association of SES with health.[8–10] It has been suggested that SES cannot directly affect health as physiological variables do, but that it acts indirectly through behavioural factors or chronic stress that determine health.[8] In addition, people of higher SES are also more likely to use knowledge, financial resources and beneficial social networks in order to evaluate risk factors and take up protective actions. On the other hand, there is also evidence for the reverse relationship between SES and health, for example, the social drift hypothesis states that poor health leads to a downward shift in social class.[11 12]

Studies that focus on the determinants of adult disease frequently ignore the influence of factors from previous stages of life. Standard practice for measuring SES in most health studies is to include measures of current socioeconomic characteristics. Even

if childhood socioeconomic position is measured, it is usually accounted for in multivariable models without taking into account the temporal relationship between variables.[13] The life course approach to how SES affects health has been increasingly applied to studying health.[14–16] The life course framework consists of several, partially interrelated models. The most frequently used concepts regarding the influence of life course SES on health are: (1) the critical period model emphasising the importance of the timing of SES exposure, (2) the social mobility model considering the influence of changes in markers of SES over time followed by upward mobility or downward mobility, which in turn may impact health, (3) the cumulative exposure model which hypothesises that factors acting at different stages of life accumulate to influence disease risk in later life.[9 17] Thus, the life course approach allows to go beyond simple measures of childhood and adulthood SES to better capture important aspects of SES exposure for disease development. While the life course approach has already been implemented to study the effects of SES on cardiovascular diseases,[18] obesity,[5] hypertension[19] and diabetes,[6] it has been applied less frequently to respiratory diseases.[20 21]

Lung function is a long-term predictor of morbidity and mortality from a range of diseases including non-respiratory conditions.[22 23] Studies have demonstrated the association between a decrease in lung function and not only increasing age, but also low SES.[24] However, little is known about how changes in SES patterns influence the respiratory health. None of the research from Central and Eastern Europe (CEE) has examined this association and hence this topic becomes challenging. This is especially because of the political transformation of 1989/1990 occurring in this region, which was followed by rapid social and economic changes, resulting in the widening of income inequalities and changes in health behaviour. Additionally, most middle-aged people's life crossed over that time period resulting in some phenomena that childhood SES was well established before and adulthood SES have been developing after the fall of socialism.

The aim of this study was to examine the socioeconomic gradient in pulmonary function in middle-aged adults. We hypothesised that social mobility is positively related to pulmonary function measured by forced vital capacity (FVC) and 1 s forced expiratory volume ($FEV_1$). The analysis was guided by three research questions, all aimed at characterising the socioeconomic patterning of lung functions after adjustment for modifiable risk factors such as smoking and body mass index (BMI): (1) whether there is a gradual (dose-related) association between childhood and adulthood SES and pulmonary function measured by FVC and $FEV_1$; (2) whether there is a relationship between social mobility and pulmonary function and (3) whether there is a relationship between cumulative SES and pulmonary function.

## METHODS
### Study design and data collection
The analysis was performed using data from the Polish part of the Health, Alcohol and Psychosocial Factors in Eastern Europe (HAPIEE) study. The design of this study has been described in detail elsewhere.[25] In summary, the study included four urban cohorts selected from population registers in Novosibirsk (Russia), Krakow (Poland), six Czech towns and Kaunas (Lithuania); each consisted of a random sample of men and women aged 45–69 years at baseline in 2002–2005. All participants completed a comprehensive health questionnaire and underwent a brief medical examination including a pulmonary function test. In Poland, trained nurses interviewed the respondents at their homes. For this analysis information about age, current and past socioeconomic circumstances, health status (asthma or history of respiratory disease), and health behaviour, that is, smoking and physical activity was obtained using a standard questionnaire. Then, all participants were invited to a clinic for anthropometric and blood pressure measurements followed by blood collection for biochemical tests and for performance of spirometry.

### Patient and public involvement statement
Participants were not involved in the design of this study.

## MEASURES
### Pulmonary function assessment
In our analysis, we used two parameters of pulmonary function: FVC, which is recognised as a predictor of vital status; and $FEV_1$, which is regarded as a good biological marker of risk of obstructive pulmonary disease. FVC and $FEV_1$ were measured by a trained technician in accordance with a standardised protocol[26] using a Micro-Medical Microplus spirometer. Participants were asked to make at least three maximal expiratory measures. If the quality of the spirometry was not satisfactory, the manoeuvre was repeated until the best quality was obtained. The highest value of FVC and the highest value of $FEV_1$ were chosen from those measurements for which the repeatability criteria were met. The quality of all tests was reviewed by a pulmonologist. Criteria for a satisfactory repeatability were based on the European Respiratory Society recommendations.[26] Exclusion criteria for the spirometry included any chest surgery during previous 3 months, any physician-diagnosed serious cardiac condition (stroke, myocardial infarction) and recent respiratory infections.

## SOCIOECONOMIC STATUS
Childhood SES was assessed using the level of education of both parents (incomplete primary or no formal education, primary, vocational, secondary or university) and information about housing standard at the age of about 10 (cold tap water, hot tap water, radio, fridge, own

kitchen and own toilet), all summed up into an index measuring the amenities in childhood.

Adulthood SES was measured by five items: the study of the participant's educational attainment, professional activity, household amenities and current financial situation. The participant's education was assessed by questioning his/her highest level of completed education (incomplete primary or no formal education, primary, vocational, secondary, or university). Current employment status encompassed the following categories: entrepreneur, employed and self-employed, unemployed including housewife and retired people. An index of household amenities was constructed by summing up the number of valuable items that a participant had in his/her household (microwave, washing machine, video recorder, colour television (TV), dishwasher, freezer, camcorder, satellite TV, telephone and mobile phone). Current financial status was assessed by two questions: 'How often does it happen not to have enough money for food which you and your family need?' and 'Do you have any difficulties with paying bills?'. The possible answers were the following: all the time, often, sometimes, rarely or never. The same indicators were used in other reports from the HAPIEE study.[27]

A two-step clustering algorithm was applied to classify participants into homogeneous groups in respect of childhood SES and adulthood SES. In our analysis, we allowed the process to identify the dimensions of SES without constraint to any particular number of factors, in order to capture the natural complexity of SES. Three groups of childhood SES were generated, assessing reasonable evidence of the cluster structure (silhouette coefficient s(i)=0.51). Participants who had poor housing conditions (low value of childhood amenities index) and whose parents had no formal education or had at most completed primary school were classified by the algorithm as a low SES group. The middle SES group comprised subjects characterised by medium housing conditions in childhood and with secondary education have been completed by at least one of the

participant's patents. A high childhood amenities index value and parent with secondary or university education classified a participant into the high childhood SES group. Similarly, three groups of adult SES were identified (s(i)=0.55). Participants who had a university education were employed, and had a high index of amenities were classified by the clustering algorithm as having high SES in adulthood. On the other hand, participants with low educational achievements, unemployed, being on a retirement pension or reporting a lack of money for food or paying bills were classified into a low SES group. The middle SES group comprised participants mainly characterised having completed secondary education (however, 44% had a primary education), half-and-half working and retired persons having no problem with money for food and paying bills regularly (table 1).

Cumulative SES score was calculated by summing the childhood and adulthood SES evaluations obtained by cluster analysis with code 0 for low SES, code 1 for medium SES and code 2 for high SES. The final cumulative SES score was further categorised as low (score=0–1), medium (score=2–3) and high (score=4).

### Transition from childhood to adulthood (social mobility)

Having separate estimates for early childhood SES and adult SES allowed each participant to be allocated to one of the following four separate groups of SES changes: participants who were classified in the low subgroups in both time points were categorised as 'always low'; those who were classified in the low SES in childhood, but who in adulthood were middle or high, were categorised as 'upward mobility'. Similarly, those classified in middle or high SES in childhood and low in adulthood were categorised as 'downward mobility'. The categories moderate and high SES were combined due to the similar differences in pulmonary function of participants who changed SES from low to moderate and from low to high SES. Finally, those classified in both life stages as having middle or high SES were categorised as 'always moderate or high'.

| Table 1 | Definition of the socioeconomic status | | |
|---|---|---|---|
| **Socioeconomic status** | | **Descriptive statistics** | **Cluster description** |
| Childhood | Low | n=1632<br>Age (years): 58.9 Female (%): 51 | ▶ Primary parent's education<br>▶ Low value on amenities index in childhood |
| | Middle | n=1463<br>Age (years): 58.4 Female (%): 51 | ▶ Primary or secondary parent's education<br>▶ Low or middle value on amenities index in childhood |
| | High | n=881<br>Age (years): 56.0 Female (%): 50 | ▶ Secondary or university parent's education<br>▶ High value on amenities index in childhood |
| Adulthood | Low | n=1313<br>Age (years): 57.8 Female (%): 58 | ▶ Primary or secondary education<br>▶ Often or sometimes do not have enough money for food or have problems with paying bills<br>▶ Unemployed or pensioner |
| | Middle | n=1588<br>Age (years): 58.9 Female (%): 49 | ▶ Primary or secondary education<br>▶ Rarely or never do not have enough insufficient money for food or have problems with paying bills |
| | High | n=1101<br>Age (years): 57.1 Female (%): 44 | ▶ University education<br>▶ Employed or owner of company<br>▶ High value on amenities index |

## STATISTICAL ANALYSIS

Descriptive statistics stratified by sex are presented as means and SD for pulmonary function parameters and other continuous measurements and as frequency (n, %) for categorical variables. Two-step cluster analysis was used to determine homogeneous subgroups of childhood SES and adult SES. The procedure is particularly designed to handle large data set and allows consideration of both continuous and categorical variables. In the first step, the algorithm uses a sequential clustering method to allocate the cases into many small subclusters. The second step takes subclusters resulting from the first step as input in order to group them into clusters. In the modelling process, the hierarchical clustering method and the log-likelihood distance measure between clusters were used. The validation of consistency within clusters of data was based on the silhouette coefficient s(i). Values above 0.5 are recognised as either reasonable or strong evidence of cluster structure (details are briefly described in ref [28]).

To compare FVC and $FEV_1$, and the distribution by SES in childhood, in adulthood, and in transition of SES between these two periods, a set of analyses of covariance were performed separately for men and women. Due to the ordinal character of the SES variables, p values for linear trends in mean values of spirometry parameters ($FEV_1$ and FVC, separately) are presented. The final models were adjusted for age, height, BMI, smoking and the prevalence of respiratory diseases. Data were analysed using IBM SPSS V.22. All tests were two tailed, and a $p<0.05$ was considered statistically significant.

## RESULTS

Out of the 4840 participants for whom spirometry measurements were performed, 4104 (84.8%) participants provided valid spirometry data and were included in this analysis. There were no differences in age, BMI, smoking and SES level in childhood between the analysed group and excluded participants. But, the latter group had a higher prevalence of respiratory diseases and a higher percentage (by 7%) of low SES category in adulthood. The characteristics of the study sample are presented in table 2. The mean age was 55.6 (SD=6.92) years; men were slightly older than women (57.9 years vs 57.2 years). Mean $FEV_1$ for the whole sample was 2.70 L (95% CI 2.68 to 2.73), men had a 0.88 L higher mean $FEV_1$ than women. Mean FVC was 3.25 L (95% CI 3.22 to 3.28), the difference between men and women in mean FVC was statistically significant and was equal to 1.08 L. The distribution of SES indicators (adulthood, cumulative and transition) differed between sexes. Low SES in adulthood was observed in 27% of men and 37% of women (p<0.001). No differences in the distribution of childhood SES emerged from sex groups.

Tables 3 and 4 show means of $FEV_1$ and FVC, related to SES indices (childhood SES, adulthood SES, cumulative SES and social mobility), in the model adjusted for age and the model adjusted for all other covariates. Age-adjusted means of $FEV_1$ and FVC were associated with childhood (in women), adulthood and cumulative SES score with a significant gradient in men and women. Compared with high SES, men in the low or middle SES had significantly worse lung function (differences ranging from 0.20 L to 0.27 L in $FEV_1$ and 0.10 L to 0.26 L in FVC). Results were similar but less pronounced in women (differences from 0.07 L to 0.16 L in case of $FEV_1$ and 0.18 L to 0.25 L in FVC). Controlling for age, height, BMI, smoking and the prevalence of respiratory diseases resulted in some attenuation of the estimates, but the pattern and significance of the association remained, except for the relationship between childhood SES and FVC, as well as cumulative SES and $FEV_1$ and FVC in men.

Social mobility was related to both $FEV_1$ and FVC. In men and women, participants whose SES decreased to low had pulmonary function similar to participants who always had low SES. Compared with persons with life course low SES or SES decreased to low, persons with life course moderate or high SES or increased SES, had higher FVC and $FEV_1$. The differences between participants who always had low SES compared with those with moderate or high SES were 190 mL for FVC and 130 mL for $FEV_1$, in men. In women, the differences were smaller and equal to 70 mL for both FVC and $FEV_1$ (tables 3–4).

## DISCUSSION

The study showed that moderate or high childhood and adulthood SES, as well as upward social mobility, were associated with a better pulmonary function, that is, higher means of $FEV_1$ and FVC. The relationship was attenuated slightly, particularly in men after adjustment for age, height, smoking, obesity and history of respiratory diseases but remained statistically significant. The attenuation was mostly explained by height, which itself serves as a marker of early-life disadvantage and directly affects the lung function.[29–31]

Results from our study, with reference to life stages of childhood or adulthood SES, are consistent with previous studies,[21 30–34] which have shown that pulmonary function is gradually worse in the lower SES groups, although the magnitude of the effect varies across studies.

In this study, we applied a novel approach to determine childhood SES and adulthood SES from the range of socioeconomic indicators. Almost all previous studies used either indicator of SES (education, income or occupation) or attempted to explain the independent effect of one indicator, by adjusting for other indices in regression analyses. However, this approach is questioned as it does not take into account the joint effect of these factors and their interrelationship.[16] In our study, we used a model-based framework (cluster analysis) to identify SES groups that captured a detailed heterogeneity in education, previous and current economic situation, and current occupational status. Cluster analysis is not a traditional approach to social stratification, although it

**Table 2** Characteristics of the studied sample by sex

| | Male n=2020 | | Female n=2084 | | |
|---|---|---|---|---|---|
| | **Mean** | **SD** | **Mean** | **SD** | **P value** |
| Age | 57.9 | 6.86 | 57.2 | 6.962 | 0.001* |
| BMI | 28.10 | 4.130 | 28.47 | 5.040 | 0.016* |
| FEV$_1$ (L) | 3.15 | 0.659 | 2.27 | 0.473 | <0.001* |
| FVC (L) | 3.81 | 0.717 | 2.73 | 5.280 | <0.001* |
| | **n** | **%** | **n** | **%** | |
| Respiratory diseases (yes) | 266 | 13.4 | 285 | 13.8 | 0.6† |
| Smoking status | | | | | |
| Current smoker | 666 | 33.1 | 539 | 25.9 | |
| Ex-smoker | 755 | 37.5 | 419 | 20.2 | <0.001† |
| Never smoker | 592 | 29.4 | 1121 | 53.9 | |
| Adulthood SES | | | | | |
| Low | 548 | 27.1 | 765 | 36.7 | |
| Middle | 804 | 39.8 | 784 | 37.6 | <0.001† |
| High | 614 | 30.4 | 487 | 23.9 | |
| Childhood SES | | | | | |
| Low | 799 | 40.7 | 833 | 41.3 | |
| Middle | 723 | 36.9 | 740 | 36.7 | 0.9† |
| High | 439 | 22.4 | 442 | 21.9 | |
| Social mobility | | | | | |
| Always low | 225 | 11.7 | 314 | 15.9 | <0.001† |
| Downward mobility | 308 | 16.1 | 424 | 21.5 | |
| Upward mobility | 557 | 29.1 | 503 | 25.5 | |
| Always moderate or high | 825 | 43.1 | 732 | 37.1 | |
| Cumulative SES score | | | | | |
| Low | 833 | 43.5 | 996 | 50.5 | |
| Middle | 729 | 43.3 | 769 | 39 | <0.001† |
| High | 253 | 13.2 | 208 | 10.5 | |

*P values for t-test.
†P values for $\chi^2$ test.
BMI, body mass index; FEV$_1$, forced expiratory volume; FVC, forced vital capacity; SES, socioeconomic status.

was applied in some studies.[35 36] This method seems to be particularly suitable for CEE society in which social classes are less clearly separated from each other. As a result, we observed that for the adult SES, low and middle SES did not differ in terms of education and home amenities index. They differed significantly in their current occupational status, as the low SES group was dominated by pensioners and unemployed people, which further, partially, explained the differences in current material resources for everyday living. The associations between the complexity of SES and pulmonary functions were in line with other studies.[30–34]

Considering childhood SES, parent's educational achievements overlap slightly between low–middle and middle–high SES groups reflecting the fact that a single indicator of education does not sufficiently capture the complex pattern of SES characteristics. However, the standard of housing (the second component of SES) separated the low and high groups unequivocally. In contrast to other studies, we did not find significant low–high SES differences in pulmonary measurements in men, after adjustments for adult risk factors. This suggests that the SES effect in childhood might be masked by SES in adulthood. Although parent's education significantly determines children's educational opportunity, during the socialist period the common national policy was to equalise the chances for education by favouring to children from lower socioeconomic groups. This policy reduced the inequality in education; however, because of rapid industrialisation of the country, education was strongly vocationally oriented. Hazardous work and the prevalence of more frequent

**Table 3** Means of FEV₁ (L) by category of childhood, adulthood SES, transition SES and cumulative assessment of SES

| | Male | | | | Female | | | |
|---|---|---|---|---|---|---|---|---|
| | Model A | | Model B | | Model A | | Model B | |
| | Mean | 95% CI | Mean | 95% CI | Mean | 95% CI | Mean | 95% CI |
| **Adulthood SES** | | | | | | | | |
| Low | 3.01 | (2.96 to 3.06) | 2.9 | (2.85 to 2.94) | 2.22 | (2.19 to 2.25) | 2.12 | (2.09 to 2.15) |
| Middle | 3.16 | (3.12 to 3.20) | 2.99 | (2.95 to 3.03) | 2.27 | (2.24 to 2.30) | 2.15 | (2.12 to 2.18) |
| High | 3.28 | (3.24 to 3.32) | 3.03 | (2.98 to 3.07) | 2.36 | (2.33 to 2.40) | 2.2 | (2.16 to 2.24) |
| P for trend | <0.001 | | <0.001 | | <0.001 | | 0.02 | |
| **Childhood SES** | | | | | | | | |
| Low | 3.15 | (3.11 to 3.19) | 3.00 | (2.95 to 3.04) | 2.27 | (2.24 to 2.29) | 2.16 | (2.13 to 2.19) |
| Middle | 3.18 | (3.08 to 3.27) | 2.95 | (2.90 to 2.99) | 2.25 | (2.22 to 2.28) | 2.12 | (2.09 to 2.16) |
| High | 3.19 | (3.14 to 3.25) | 2.94 | (2.89 to 3.00) | 2.34 | (2.30 to 2.37) | 2.19 | (2.15 to 2.23) |
| P for trend | 0.1 | | 0.1 | | 0.001 | | 0.011 | |
| **Cumulative SES score** | | | | | | | | |
| Low | 3.09 | (3.05 to 3.13) | 2.95 | (2.91 to 3.00) | 2.24 | (2.18 to 2.24) | 2.14 | (2.11 to 2.17) |
| Middle | 3.18 | (3.14 to 3.22) | 2.97 | (2.93 to 3.01) | 2.29 | (2.26 to 2.32) | 2.15 | (2.12 to 2.18) |
| High | 3.29 | (3.22 to 3.36) | 2.99 | (2.92 to 3.06) | 2.4 | (2.35 to 2.46) | 2.23 | (2.17 to 2.28) |
| P for trend | <0.001 | | 0.6 | | <0.001 | | 0.009 | |
| **Social mobility** | | | | | | | | |
| Always low | 2.98 | (2.90 to 3.05) | 2.91 | (2.84 to 2.98) | 2.21 | (2.17 to 2.26) | 2.14 | (2.10 to 2.19) |
| Downward mobility | 3.03 | (2.97 to 3.10) | 2.88 | (2.82 to 2.94) | 2.23 | (2.19 to 2.27) | 2.13 | (2.09 to 2.17) |
| Upward mobility | 3.21 | (3.16 to 3.26) | 3.03 | (2.98 to 3.08) | 2.3 | (2.27 to 2.34) | 2.19 | (2.15 to 2.23) |
| Always moderate or high | 3.21 | (3.17 to 3.24) | 3.04 | (2.98 to 3.07) | 2.31 | (2.28 to 2.34) | 2.21 | (2.18 to 2.24) |
| P for trend | <0.001 | | <0.001 | | <0.001 | | 0.048 | |

Model A—adjusted for age.
Model B—adjusted for age, height, BMI, smoking status and respiratory disease.
BMI, body mass index; FEV₁, forced expiratory volume; SES, socioeconomic status.

smoking in men may partially explain the lack of this significance.[37 38]

The present study supported two life course hypotheses originating from the social mobility and cumulative SES frameworks. We showed that social mobility was related with a functional lung state in adult life. To our knowledge, only two studies have attempted to analyse SES trajectories during life course in relation to lung function disparities. Menezes *et al* showed that SES at birth, in adolescence and its trajectory between these two periods was inversely associated with lung function measurements by FEV₁ and FVC in Brazilians.[39] Ramsay *et al* showed that combination of adverse socioeconomic position in both adult life and childhood was additionally associated with a greater decline in lung function in men.[34]

Within the cumulative SES score, our results showed that the accumulation of social disadvantage is linked to poorer lung function in women; however, we did not find this pattern in men. After adjustment for adult SES-related covariates, graded associations with pulmonary measurements were attenuated, which suggests that childhood conditions are less important in men and this supports the significance of contemporary SES-related factors.

The mechanism of the association between SES and lung function is not fully understood. The biological pathway by which limited lung development in the prenatal period is related to infant respiratory infection and higher sensitivity to impaired lung function in adulthood is relatively well established. Another explanation is the sociobiological pathway: SES in childhood is adversely associated with postnatal lung function, which leads to worse lung development as well as an impact on the immune system through increased exposure to risk factors such as air pollution, passive smoking or poor nutrition. Respiratory diseases in childhood may be an obstacle in gaining the right education, which leads to lower SES in adulthood.

Differences in respiratory system functioning among adults with certain SES may be explained by differences in occupation especially the intensity and frequency of physically hazardous work that has a direct impact on the respiratory system.[34] In addition, low SES is associated with more frequent exposure to risk factors, such as nutrition disorders, smoking, more frequent infections and obstructive pulmonary disease.[37 40–42]

**Table 4** Means of FVC (L) by category of childhood, adulthood, transition SES and cumulative assessment of SES

| | Male | | | | Female | | | |
|---|---|---|---|---|---|---|---|---|
| | Model A | | Model B | | Model A | | Model B | |
| | Mean | 95% CI | Mean | 95% CI | Mean | 95% CI | Mean | 95% CI |
| Adulthood SES | | | | | | | | |
| Low | 3.66 | (3.61 to 3.72) | 3.61 | (3.55 to 3.67) | 2.67 | (2.64 to 2.71) | 2.63 | (2.59 to 3.66) |
| Middle | 3.83 | (3.78 to 3.88) | 3.73 | (3.68 to 3.78) | 2.71 | (2.67 to 2.74) | 2.65 | (2.61 to 2.69) |
| High | 3.92 | (3.86 to 3.97) | 3.71 | (3.65 to 3.76) | 2.85 | (2.81 to 2.89) | 2.73 | (2.68 to 2.77) |
| P for trend | <0.001 | | 0.005 | | <0.001 | | <0.001 | |
| Childhood SES | | | | | | | | |
| Low | 3.79 | (3.74 to 3.83) | 3.71 | (3.66 to 3.76) | 2.69 | (2.66 to 2.72) | 2.642 | (2.61 to 2.68) |
| Middle | 3.77 | (3.72 to 3.82) | 3.66 | (3.60 to 3.71) | 2.71 | (2.68 to 2.75) | 2.635 | (2.60 to 2.67) |
| High | 3.89 | (3.83 to 3.96) | 3.69 | (3.62 to 3.75) | 2.84 | (2.79 to 2.88) | 2.725 | (2.68 to 2.77) |
| P for trend | 0.001 | | 0.1 | | <0.001 | | 0.001 | |
| Cumulative SES score | | | | | | | | |
| Low | 3.73 | (3.67 to 3.78) | 3.67 | (3.62 to 3.72) | 2.67 | (2.64 to 2.70) | 2.63 | (2.60 to 2.67) |
| Middle | 3.84 | (3.80 to 3.89) | 3.69 | (3.64 to 3.74) | 2.76 | (2.72 to 2.79) | 2.66 | (2.62 to 2.70) |
| High | 3.93 | (3.84 to 4.02) | 3.67 | (3.58 to 3.75) | 2.92 | (2.85 to 2.98) | 2.78 | (2.71 to 2.84) |
| P for trend | <0.001 | | 0.7 | | <0.001 | | <0.01 | |
| Social mobility | | | | | | | | |
| Always low | 3.59 | (3.50 to 3.68) | 3.6 | (3.52 to 3.67) | 2.64 | (2.58 to 2.69) | 2.62 | (2.57 to 2.67) |
| Downward mobility | 3.72 | (3.64 to 3.80) | 3.63 | (3.56 to 3.70) | 2.71 | (2.66 to 2.75) | 2.64 | (2.59 to 2.68) |
| Upward mobility | 3.87 | (3.81 to 3.92) | 3.77 | (3.71 to 3.83) | 2.72 | (2.68 to 2.76) | 2.67 | (2.63 to 2.72) |
| Always moderate or high | 3.86 | (3.81 to 3.91) | 3.79 | (3.74 to 3.85) | 2.8 | (2.76 to 2.83) | 2.69 | (2.66 to 2.73) |
| P for trend | <0.001 | | <0.001 | | <0.001 | | 0.02 | |

Model A—adjusted for age.
Model B—adjusted for age, height, BMI, smoking status and respiratory disease.
BMI, body mass index; FVC, forced vital capacity; SES, socioeconomic status.

All the above factors are simultaneously related with the SES. That is why the most possible explanation for the relationship between SES and the deterioration of respiratory functions is multifactorial due to the simultaneous influence of many factors. This would enable an explanation, a partial one at least, of the described relationships.

Our study has introduced some new aspects to current knowledge. First, this study is the first in CEE which considers SES in the light of social and political transformations after the fall of the communist system. Second, we have defined model-based measures of childhood SES and adulthood SES that capture the multidimensionality of SES, which provide a detailed picture of the defined SES levels. This novel approach might facilitate the understanding of the complexity of the SES construct. For example, while about 60% of men from the high SES childhood group had a university education, which would place them in higher the SES group when taking education as a single SES indicator, this percentage dropped to 47% for those belonging to the high SES group, when assessed as a composite measure. This highlighted the inconsistency in SES groups, which can be explained in two ways. First, levels of education do not fit together with other dimensions of SES. Second, the current economic situation (one of the elements of our SES composite measure) that provides resources to control one's circumstances overcomes the educational level among people who are nearly 60 years old and live in this part of Europe.

However, there are some limitations to the interpretation of the results. First, the cross-sectional study design does not allow address the causality of the observed relations. SES and especially its economical components are sensitive to changes over time. This may influence the health and then support health selection hypothesis, which in turn would strengthen the relationship between SES and the studied health outcome. Second, information on childhood SES was based on the distant recall of various markers of social circumstances (up to 59 years in the oldest study participants). The retrospective nature of this of information increases the random error and it is subject to recall bias. The known effect of the overestimation of socioeconomic position in childhood, assessed on the basis of parental education, might also bias the results.[43] Finally, broader generalisation of the results is limited as the cluster analysis which was used to assess the SES categories is sensitive to the nature of the data, types of relationships between variables and the methods of grouping cases into distinct clusters.

## CONCLUSION

Low SES over a life course was associated with the lowest lung function. Downward social mobility was associated with a poorer pulmonary function while upward mobility or life course moderate or high SES was associated with a better pulmonary function.

**Contributors** MP drafted the manuscript, designed the analysis and performed the data analysis. KS contributed to the data analysis and helped draft the manuscript, critical revised the manuscript. MK was responsible for critical revision of the manuscript. RW-W was responsible for tables preparations and critical revision of the manuscript. MB contributed to the study design and critical revision of the manuscript. AP contributed to the study design and critical revision of the manuscript. All authors approved the final version of the manuscript and this submission.

**Funding** The authors have disclosed the receipt of the following financial support for the research, authorship and/or publication of this article: this work was funded by the Wellcome Trust (grant WT064947 and WT081081), the US National Institute of Aging (grant R01 AG23522) and the MacArthur Foundation.

**Competing interests** All the authors report grants from The Wellcome Trust, grants from the US National Institute of Aging, grants from The MacArthur Foundation during the conduct of the study. AP: personal fees from AMGEN and Sanofi, not related with the submitted work.

**Patient consent for publication** Not required.

**Ethics approval** The study was approved by the Bioethics Committee of the Jagiellonian University and by the University College London/University College London Hospital ethics committee.

**Provenance and peer review** Not commissioned; externally peer reviewed.

**Data sharing statement** Data not in public domain.

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
