## [Reviewer comments · BMJ Open]

ARTICLE DETAILS

TITLE (PROVISIONAL)	Socioeconomic status and pulmonary function, transition from childhood to adulthood: cross-sectional results from the Polish part of the HAPIEE study.
AUTHORS	Polak, Maciej; Szafraniec, Krystyna; Kozela, Magdalena; Wolfshaut-Wolak, Renata; Bobak, Martin; Pajak, Andrzej

VERSION 1 – REVIEW

REVIEWER	Gabriela Oates, PhD University of Alabama at Birmingham, USA
REVIEW RETURNED	12-Apr-2018

GENERAL COMMENTS	This is an important article in a critical area of public health research that has not received the attention it deserves: the role of social mobility in pulmonary health. The study has several strengths, including a large population sample. Below are some comments and suggestions that can help strengthen the manuscript. 1. Introduction It is important to introduce the theoretical framework that informs this study in order to justify the selection of study measures. Specifically, I recommend that the authors briefly present (1) the multidimensional concept of SES, (2) the concept of social mobility, (3) conceptual frameworks that explain the effect of SES on health and are relevant to the study, and (4) the life course approach, with appropriate citations for topic each. For example, the authors use a cumulative SES measure without introducing the concept of cumulative disadvantage; its first mention is in the second paragraph of the discussion. On the other hand, they introduce the concept of allostatic load, yet the study does not use measures of allostatic load. Similarly, the concept of perceived or subjective social position is introduced without actual follow-up in the study. Second, any hypotheses proposed by the authors need to be stated at the end of the introduction, not in the discussion. Relatedly, the statistical analysis section needs to explain how the hypotheses were tested, while the results and discussion sections need to state whether the hypotheses were supported or not. Third, it is important to use consistent terminology through the paper. For example, the authors seem to use interchangeably terms such as 'social mobility,' 'transition(s) SES,' and 'change(s) in SES.' When it comes to SES, there also seems to be a
--

	confusion between index, variable, and indicator or aspect and what they mean. Why is the opening paragraph focusing on CVD in particular? There is a wealth of evidence for the effect of SES on a range of health outcomes that are not condition-specific. Literature that is 10 to 20 years old cannot be referred to as “recent.” 2. Methods I would recommend adding a subtitle ‘Measures’ under which all measures used in the study are described. Childhood SES is measured with 2 items: parental education (in 5 categories) and housing standards (range 0-6), while adulthood SES is measured with four items: education (the categories are not provided), professional activity (owner, employed, retired, unemployed), household amenities (range 0-12), and financial situation (assessed with 2 questions about ability to buy food and pay bills, each on a scale of 1-5). I have several methodological questions and concerns.  • Were any of the measures standardized? • Were any weights applied? • Was the education of each parent counter separately in the childhood SES? • What are the categories of educational attainment in adulthood? • Why these specific 6 and 12 household amenities in childhood and adulthood? E.g. who decides what is considered “valuable” in a participants household? Have these measures been used in prior studies? Any references will be very helpful. • The measure ‘professional activity,’ also referred to as ‘professional position’ (please use only one term) really measures employment status: employed, unemployed, or retired, plus an additional category for ‘owner of a company.’ Has this measure been used previously? The measure doesn’t tell us anything about the occupational or professional status of a participant, although it does seem to assume that ‘owner of a company’ is more prestigious than ‘employed’ and that ‘retired’ is the least prestigious of the three. Can the authors provide evidence in support of such categorization as a measure of status in the social hierarchy? • Cumulative SES: (1) Please introduce and explain the cumulative SES measure in a separate paragraph as done for childhood SES and adulthood SES. Also, as stated previously, the concept of cumulative SES needs to be part of the introduction to justify the use of the measure. (2) Calculating cumulative SES: childhood and adulthood SES used different measures. Can they just be summed up to be an index? Were the measures standardized or were there any weights applied? Cluster analysis is a non-traditional approach to social stratification. It is highly dependent on the sample and on what measures are included. It would be very helpful if the authors can justify their approach with examples of use from the sociological literature or in prior health studies. “Transition from childhood to adulthood” subtitle: I would recommend a consistent use of the term “social mobility”
--	---

(downward or upward) in place of various uses of 'transition' or 'change.'

Please include language to justify why both FEV1 and FVC are used, and what is the significance of including both measures. Please address this in the discussion as well.

When stating the significance level please also state if the tests were two-tailed.

3. Results

Table 2:

- Sample size should be placed below Male or Female, and the column currently labeled with the sample size should show the appropriate label. (Is that standard deviation? Unclear.)
- Please add % to the categorical variables.
- For readability, please use indent for the subcategories under a category
- Please use consistent terminology across tables (e.g., T2 uses 'Transitions SES' while T3 uses 'Change in SES')

Tables 3 and 4:

- Only one p-value is provided for variables that have multiple categories. E.g., adult SES has low, middle, and high categories. What is the p-value for? Is there a reference category? Do the authors compare low to high, low to middle, and middle to high? Unclear.
- It is stated that Model A adjusts for age, while Model B adjusts for additional four variables. How was the adjustment made? The tables present means and look like bivariate tables of FEV1/FVC and SES. Can the authors justify why they are not using regression analysis (assuming they are not by the absence of regression coefficient)?
- Page 10, line 19-22: The interpretation is not straightforward and readers have to calculate the differences in lung function.

4. Discussion

As the discussion typically opens with a summary of the study, from the most to least important, I recommend using the second paragraph as an opening paragraph for this section.

Page 13, line 17-19: It is not clear why this 'novel approach' is a good measure of SES. The authors did not cite previous work and did not justify why those specific variables were used in the cluster analysis.

Page 13, line 33-36: There seems to be a confusion about the meaning of index and variable. SES indices (e.g., Duncan, Hollingshead, Nam-Powers, or the MacArthur scale) include several dimensions (e.g., variables such as education, income, and occupation) that indicate one's position in the social hierarchy. Multiple studies assess SES by more than one indicator and measure access to a variety of financial, human, and social resources. There is also a wealth of evidence, both in the sociological and the general medical literature, that various SES indicators are differently associated with various health outcomes (e.g. Braveman et al. SES in health research: one size does not fit all. JAMA 2005). In the opinion of this reviewer, the effect of such indicators on health, including their interactions, can be studied

	effectively with various statistical methods, including regression analysis. Conversely, cluster analysis cannot compensate for poor conceptualization or for the lack of measures that include all dimensions of the SES concept. Page 13, last paragraph: Childhood SES and pulmonary measurements were not significant in men, but significant in women. Could the authors comment on that? Lastly, I encourage the authors to seek editorial services as the manuscript does not conform to the conventions of standard English usage and grammar.
REVIEWER	Timothy Ore Department of Health and Human Services, 50 Lonsdale Street, Melbourne, AUSTRALIA 3000
REVIEW RETURNED	30-Apr-2018
GENERAL COMMENTS	Would be useful to have more than three groups of childhood SES; I suggest quartiles or quintiles, as these will provide results with greater levels of sensitivity. If possible, adjustments (in the models) for the presence of key non-respiratory chronic conditions will be of great help to advancing knowledge in this important area. Findings are entirely tentative, pending corroboration from more reliable designs, in particular, longitudinal cohorts. Is there a way to include variables for smoking duration? Overall, a good paper.

VERSION 1 – AUTHOR RESPONSE

Reviewer: 1

Reviewer Name: Gabriela Oates, PhD

Dear doctor Gabriela Oates,

> Please state any competing interests or state “None declared”

The appropriate information is given at the end of a paper (marked in red, page 19, section 3)

> This is an important article in a critical area of public health research that has not received the attention it deserves: the role of social mobility in pulmonary health. The study has several strengths, including a large population sample. Below are some comments and suggestions that can help strengthen the manuscript.

We do appreciate this comment very much.

> 1. Introduction

It is important to introduce the theoretical framework that informs this study in order to justify the selection of study measures. Specifically, I recommend that the authors briefly present (1) the multidimensional concept of SES, (2) the concept of social mobility, (3) conceptual frameworks that explain the effect of SES on health and are relevant to the study, and (4) the life course approach,

with appropriate citations for topic each. For example, the authors use a cumulative SES measure without introducing the concept of cumulative disadvantage; its first mention is in the second paragraph of the discussion. On the other hand, they introduce the concept of allostatic load, yet the study does not use measures of allostatic load. Similarly, the concept of perceived or subjective social position is introduced without actual follow-up in the study.

To meet the suggestions we re-wrote the introduction (marked in red).

> Second, any hypotheses proposed by the authors need to be stated at the end of the introduction, not in the discussion. Relatedly, the statistical analysis section needs to explain how the hypotheses were tested, while the results and discussion sections need to state whether the hypotheses were supported or not.

Thank you for the suggestion, appropriate text was added at the end of introduction (page 5, section 2).

> Third, it is important to use consistent terminology through the paper. For example, the authors seem to use interchangeably terms such as 'social mobility,' 'transition(s) SES,' and 'change(s) in SES.' When it comes to SES, there also seems to be a confusion between index, variable, and indicator or aspect and what they mean.

Terminology was unified.

> Why is the opening paragraph focusing on CVD in particular? There is a wealth of evidence for the effect of SES on a range of health outcomes that are not condition-specific.

Introduction was re-written and the controversial phrase was not used.

> Literature that is 10 to 20 years old cannot be referred to as "recent."

Introduction was re-written and this phrase was deleted.

> 2. Methods

> I would recommend adding a subtitle 'Measures' under which all measures used in the study are described.

The subtitle "Measures" was added (page 6).

> Childhood SES is measured with 2 items: parental education (in 5 categories) and housing standards (range 0-6), while adulthood SES is measured with four items: education (the categories are not provided), professional activity (owner, employed, retired, unemployed), household amenities (range 0-12), and financial situation (assessed with 2 questions about ability to buy food and pay bills, each on a scale of 1-5). I have several methodological questions and concerns.

- > • Were any of the measures standardized?
- > • Were any weights applied?

We used two-step cluster analysis, which allows to deal with categorical and continuous variables, and to take into account the multidimensional character of SES. This approach does

not use traditional Euclidian measure of distance between two clusters but measure it as corresponding decrease in log-likelihood by combining together. For this reasons we did not apply weights or standardization of measures.

> • Was the education of each parent counter separately in the childhood SES?

We have taken information of education of both parents. Information was provided in the methods section (page 7, section). Text marked in red

> • What are the categories of educational attainment in adulthood?

We added information about categories of participant's education into the section on methods (page 7, section 3). Text marked in red

> • Why these specific 6 and 12 household amenities in childhood and adulthood? E.g. who decides what is considered "valuable" in a participants household? Have these measures been used in prior studies? Any references will be very helpful.

We believe that the list of household amenities used as marker of material circumstances seems to be relevant for CEE population. This was used in the other reports. We added information in method section and provided references (page 7, section 3).

> • The measure 'professional activity,' also referred to as 'professional position' (please use only one term) really measures employment status: employed, unemployed, or retired, plus an additional category for 'owner of a company.' Has this measure been used previously? The measure doesn't tell us anything about the occupational or professional status of a participant, although it does seem to assume that 'owner of a company' is more prestigious than 'employed' and that 'retired' is the least prestigious of th three. Can the authors provide evidence in support of such categorization as a measure of status in the social hierarchy?

Thank you for the important comment, we changed the phrase to "current employment status" to reflect the trait better.

> • Cumulative SES: (1) Please introduce and explain the cumulative SES measure in a separate paragraph as done for childhood SES and adulthood SES. Also, as stated previously, the concept of cumulative SES needs to be part of the introduction to justify the use of the measure. (2) Calculating cumulative SES: childhood and adulthood SES used different measures. Can they just be summed up to be an index? Were the measures standardized or were there any weights applied?

Thank you for this suggestion. The definition of cumulative SES was explained in a separate paragraph. The categories of SES in adulthood and childhood were obtained by cluster analysis and they are independent of the units of the measured traits. As a consequence, they could be summed up.

> Cluster analysis is a non-traditional approach to social stratification. It is highly dependent on the sample and on what measures are included. It would be very helpful if the authors can justify their approach with examples of use from the sociological literature or in prior health studies.

Thank you for this comment. In the discussion we added the text: "Cluster analysis is not a traditional approach to social stratification, although it was applied in some studies [35-36]". This methods seems to be particularly suitable for CEE society in which social classes are less clearly separated from each other (page 15, section 3). Text marked in red

> "Transition from childhood to adulthood" subtitle: I would recommend a consistent use of the term "social mobility" (downward or upward) in place of various uses of 'transition' or 'change.'

Thank you for this suggestion, we unified terminology to "social mobility".

> Please include language to justify why both FEV1 and FVC are used, and what is the significance of including both measures. Please address this in the discussion as well.

Explanatory remarks were added in the method section (page 6, section 3). Text marked in red

> When stating the significance level please also state if the tests were two-tailed.

Information was added (page 9, section 3). Text marked in red

> 3. Results

>

> Table 2:

> • Sample size should be placed below Male or Female, and the column currently labeled with the sample size should show the appropriate label. (Is that standard deviation? Unclear.)

> • Please add % to the categorical variables.

> • For readability, please use indent for the subcategories under a category

> • Please use consistent terminology across tables (e.g., T2 uses 'Transitions SES' while T3 uses 'Change in SES')

Table 2 was modified to meet the above suggestions.

> Tables 3 and 4:

> • Only one p-value is provided for variables that have multiple categories. E.g., adult SES has low, middle, and high categories. What is the p-value for? Is there a reference category? Do the authors compare low to high, low to middle, and middle to high? Unclear

In the section methods we added explanatory sentence: "Due to ordinal character of the SES variables, p-values for linear trends in mean values of spirometry parameters (FEV1 and FVC, separately) are presented". (Page 9, section 3). Text marked in red

> • It is stated that Model A adjusts for age, while Model B adjusts for additional four variables. How was the adjustment made? The tables present means and look like bivariate tables of FEV1/FVC and SES. Can the authors justify why they are not using regression analysis (assuming they are not by the absence of regression coefficient)?

We used Analysis of Covariance (ANCOVA), which is a kind of the linear regression model and we think it is more appropriate when we want to compare the differences in means between categories of independent variable. This method is multivariate model, which allows to adjust for covariates. We build two models, model 1 - with SES categories and age only, and model 2 - with SES categories, age, height, smoking status, respiratory disease and BMI. The analysis were done separately for FEV1 and FVC.

> • Page 10, line 19-22: The interpretation is not straightforward and readers have to calculate the differences in lung function.

The text was modified and the differences were added (page 14, section 1). Text marked in red

> 4. Discussion

> As the discussion typically opens with a summary of the study, from the most to least important, I recommend using the second paragraph as an opening paragraph for this section.

Thank you for this suggestion. The order of the paragraphs was changed.

> Page 13, line 17-19: It is not clear why this ‘novel approach’ is a good measure of SES. The authors did not cite previous work and did not justify why those specific variables were used in the cluster analysis.

The term “novel” was deleted.

> Page 13, line 33-36: There seems to be a confusion about the meaning of index and variable. SES indices (e.g., Duncan, Hollingshead, Nam-Powers, or the MacArthur scale) include several dimensions (e.g., variables such as education, income, and occupation) that indicate one’s position in the social hierarchy. Multiple studies assess SES by more than one indicator and measure access to a variety of financial, human, and social resources. There is also a wealth of evidence, both in the sociological and the general medical literature, that various SES indicators are differently associated with various health outcomes (e.g. Braveman et al. SES in health research: one size does not fit all. JAMA 2005). In the opinion of this reviewer, the effect of such indicators on health, including their interactions, can be studied effectively with various statistical methods, including regression analysis. Conversely, cluster analysis cannot compensate for poor conceptualization or for the lack of measures that include all dimensions of the SES concept.

We agree with the reviewer on limitations of cluster analysis. However, we have chosen it because of reasons were mentioned above. Further, indicators of SES are strongly correlated and application of linear regression could not be relevant (due to multicollinearity). For this reasons we believe that cluster analysis is meets better the specificity of our data.

> Page 13, last paragraph: Childhood SES and pulmonary measurements were not significant in men, but significant in women. Could the authors comment on that?

In the discussion section we listed potential reasons that the association between childhood SES and pulmonary function were not significant in men:

“In contrast to other studies, we did not find significant low–high SES differences in pulmonary measurements in men, after adjustments for adult risk factors. This suggests that the SES effect in childhood might be masked by SES in adulthood. Though parent’s education significantly determines children’s educational opportunity, during the socialist times the common national policy was to equal the chances for education by giving favors to children from lower socioeconomic groups. This policy reduced the inequality in education; however, because of quick industrialization of the country, education was strongly oriented to vocational level. Hazardous work and more frequent smoking in men may partially explain the lack of the significant finding].”

> Lastly, I encourage the authors to seek editorial services as the manuscript does not conform to the conventions of standard English usage and grammar.

The text was reviewed by the professional linguistic company.

Please state any competing interests or state 'None declared': None

The appropriate information is given at the end of a paper (marked in red, page 19, section 3)

> Please leave your comments for the authors below

> Would be useful to have more than three groups of childhood SES; I suggest quartiles or quintiles, as these will provide results with greater levels of sensitivity. If possible, adjustments (in the models) for the presence of key non-respiratory chronic conditions will be of great help to advancing knowledge in this important area. Findings are entirely tentative, pending corroboration from more reliable designs, in particular, longitudinal cohorts. Is there a way to include variables for smoking duration? Overall, a good paper.

We would like to thank for general opinion. Concerning the suggestion on alternative SES categories, we used two-step cluster analysis and the number of clusters is determined by the algorithm. We dealt with a problem of the conditions affecting pulmonary function by restriction of the study group (page). "Exclusion criteria for the spirometry included any chest surgery during last three months, any physician-diagnosed serious cardiac condition (stroke, myocardial infarction), and recent respiratory infections". We performed the analysis using the variable number of years of smoking. The results wed very similar. As the information on smoking status is more complete and more reliable we believe, we would prefer to leave our results as in original version.

VERSION 2 – REVIEW

REVIEWER	Gabriela Oates, PhD University of Alabama at Birmingham, USA
REVIEW RETURNED	07-Jul-2018

GENERAL COMMENTS	The authors have addressed all comments and have made the suggested changes. Several minor concerns remain: 1. Abstract The abstract was not revised to reflect the changes made to the rest of the manuscript (e.g., "professional activity" was changed to "employment status" in the main text but not in the abstract; similarly, upward and downward social mobility were used in the main text, but the Results section of the abstract retains the older phraseology).2. Strengths and limitations
---

	I recommend limiting the last bullet to the multidimensionality of the SES. Whether the use of cluster analysis is a strength is debatable. 3. Introduction Page 5, last paragraph: The authors have improved the paper by including the research questions. They may also want to state the study hypotheses. 4. Discussion The first sentence belongs to the third paragraph, which addressed the multidimensionality of SES and the use of cluster analysis. The opening paragraph of the Discussion should summarize the study results, including the direction of the relationship. A note on English usage and grammar I strongly advise the authors to seek professional editing services. Although my responsibility as a reviewer is not to copyedit the paper, it may be helpful to point a few instances of incorrect grammar, usage, or typos that limit the manuscript's readability:  • Page 4, The first sentence is ungrammatical. It could be fixed by removing "persisted." • Page 4, line 15: "conceptualized by combination of..." Mostly likely the authors mean "conceptualized to include..." • Page 4, line 30: Instead of "advocated," one of many better word choices is "suggested." • Page 4, second paragraph, last sentence: Suggested edit, "On the other hand, there is also evidence for a reverse relationship between SES and health: e.g., the social drift hypothesis states that poor health leads to a downward shift in social class." • Page 4, last line: "heath" should be "health" • Page 12, line 17: "adjusted to" should be "adjusted for" • Page 16, first sentence: Suggested edit, "In this study, we applied a novel approach to determine childhood SES and adulthood SES from a range of socioeconomic indicators." • Page 17, lines 24-25: Suggested edit, "The study supported two life-course hypotheses originating from the social mobility and cumulative SES frameworks." • Page 17, line 29: should be "only two studies have" • Use Oxford comma (e.g., page 12, line 16, text in parenthesis, but the recommendation applies to the entire text)
--	---

REVIEWER	Timothy Ore Department of Health and Human Services, Victoria, AUSTRALIA
REVIEW RETURNED	26-Jun-2018
GENERAL COMMENTS	Manuscript quality has improved, but study limitations could be elaborated further. Some editorial work will be necessary.

VERSION 2 – AUTHOR RESPONSE

Reviewer: 2

Please leave your comments for the authors below

Manuscript quality has improved, but study limitations could be elaborated further. Some editorial work will be necessary.

***In the discussion section we added a sentence about limitation of using cluster methods:
“Finally, broader generalization of the results is limited as the cluster analysis which was used to assess the SES categories is sensitive to the nature of the data, types of relationships between variables and the methods of grouping cases into distinct clusters”***

1. Abstract

The abstract was not revised to reflect the changes made to the rest of the manuscript (e.g., “professional activity” was changed to “employment status” in the main text but not in the abstract; similarly, upward and downward social mobility were used in the main text, but the Results section of the abstract retains the older phraseology).

The abstract was revised and the phrases mentioned in the comment were changed.

2. Strengths and limitations

I recommend limiting the last bullet to the multidimensionality of the SES. Whether the use of cluster analysis is a strength is debatable.

The item was removed.

3. Introduction

Page 5, last paragraph: The authors have improved the paper by including the research questions. They may also want to state the study hypotheses.

We added a sentence to the last paragraph on page 5:

“We hypothesized that social mobility is positively related to pulmonary function measured by forced vital capacity (FVC) and one-second forced expiratory volume (FEV1)”

4. Discussion

The first sentence belongs to the third paragraph, which addressed the multidimensionality of SES and the use of cluster analysis. The opening paragraph of the Discussion should summarize the study results, including the direction of the relationship.

The sentence was moved to the third paragraph and a paragraph summarizing the results was slightly re-formulated.

5. A note on English usage and grammar

I strongly advise the authors to seek professional editing services. Although my responsibility as a reviewer is not to copyedit the paper, it may be helpful to point a few instances of incorrect grammar, usage, or typos that limit the manuscript's readability:

- Page 4, The first sentence is ungrammatical. It could be fixed by removing "persisted."
- Page 4, line 15: "conceptualized by combination of..." Mostly likely the authors mean "conceptualized to include..."
- Page 4, line 30: Instead of "advocated," one of many better word choices is "suggested."
- Page 4, second paragraph, last sentence: Suggested edit, "On the other hand, there is also evidence for a reverse relationship between SES and health: e.g., the social drift hypothesis states that poor health leads to a downward shift in social class."
- Page 4, last line: "heath" should be "health"
- Page 12, line 17: "adjusted to" should be "adjusted for"
- Page 16, first sentence: Suggested edit, "In this study, we applied a novel approach to determine childhood SES and adulthood SES from a range of socioeconomic indicators."
- Page 17, lines 24-25: Suggested edit, "The study supported two life-course hypotheses originating from the social mobility and cumulative SES frameworks."
- Page 17, line 29: should be "only two studies have"
- Use Oxford comma (e.g., page 12, line 16, text in parenthesis, but the recommendation applies to the entire text)

We do appreciate the reviewers help. All corrections suggested were entered. Further, the entire text was revised by the professional linguistic company.